# Effects of different educational interventions on cervical cancer knowledge and human papillomavirus vaccination uptake among young women in Japan: Preliminary results of a cluster randomized controlled trial

Yuko Takahashi[1], Yukifumi Sasamori[1], Risa Higuchi[1], Asumi Kaku[1], Tomoo Kumagai[1], Saya Watanabe[1], Miki Nishizawa[1], Kazuki Takasaki[1], Haruka Nishida[1], Takayuki Ichinose[1], Mana Hirano[1], Yuko Miyagawa[1], Haruko Hiraike[1], Koichiro Kido[1], Hirono Ishikawa[2], Kazunori Nagasaka[1]*

1 Department of Obstetrics and Gynecology, Teikyo University School of Medicine, Tokyo, Japan,
2 Graduate School of Public Health, Teikyo University, Tokyo, Japan

* nagasakak@med.teikyo-u.ac.jp

## Abstract

The incidence and mortality rates of cervical cancer are increasing among young Japanese women. In November 2021, the Japanese Ministry of Health, Labour, and Welfare reinstated the active recommendation of the human papillomavirus (HPV) vaccine, after it had been suspended in June 2013 due to reports of adverse reactions. However, vaccine hesitancy is prevalent in the younger generation in Japan. To identify obstacles to vaccine uptake, we conducted a randomized study using different methods to provide educational content to improve health literacy regarding cervical cancer and HPV vaccination among Japanese female students. We surveyed 188 Japanese female students, divided into three groups according to the intervention: no intervention, print-based intervention, and social networking service-based intervention. Twenty questionnaires and the Communicative and Critical Health Literacy scales were used as health literacy scales. Participants' knowledge and health literacy improved regardless of the method of education. In fact, participants acquired proper knowledge when given the opportunity to learn about the importance of the disease and its prevention. Therefore, medical professionals in Japan must provide accurate scientific knowledge regarding routine HPV vaccination and the risk of cervical cancer in young women to improve their health literacy and subsequently increase HPV vaccination rates in Japan, which may lead to cervical cancer elimination.

**Trial registration number**: UMIN000036636.

## Introduction

Cervical cancer is the fourth most common cancer among women globally, with an estimated 604,000 new cases and 342,000 deaths reported in 2020 [1]. Cervical cancer is the 10th most

**Data Availability Statement:** All relevant data are within the article and its Supporting information files.

**Funding:** This research was funded in part by the Investigator-Initiated Studies Program of Merck Sharp & Dohme Corp. (Kenilworth, NJ, USA) and MSD K.K. (grant number 58246). The opinions expressed in this study are those of the authors and do not necessarily represent those of Merck Sharp & Dohme Corp. or MSD K.K. The funders had no role in study design, data collection and analysis, decision to publish, or preparation of the manuscript. No other funding or sources of support were received during this study. There was no additional external funding received for this study.

**Competing interests:** I have read the journal's policy and the authors of this manuscript have the following competing interests: This research was funded in part by the Investigator-Initiated Studies Program of Merck Sharp & Dohme Corp. (Kenilworth, NJ, USA) and MSD K.K. (grant number 58246). The opinions expressed in this study are those of the authors and do not necessarily represent those of Merck Sharp & Dohme Corp. or MSD K.K. This does not alter our adherence to PLOS ONE policies on sharing data and materials.

common cancer among women in Japan and the second most common among women aged 15–44 years [2]. Human papillomavirus (HPV) causes cervical as well as vaginal, vulvar, and head and neck cancers [3–7].

The Japanese government approved the HPV vaccine in October 2009. Public funding was provided for HPV vaccination in April 2013, targeting students from the sixth grade of elementary school to the first year of high school (approximately aged 11–16 years, in accordance with the World Health Organization recommendation) as a project to promote emergency measures for vaccination against cervical cancer [8]. However, the government withdrew its active recommendation for the vaccine on June 14, 2013, due to reports of post-vaccination adverse reactions, such as chronic pain and motor dysfunction [9]. Consequently, although the HPV vaccination rate for girls born between 1994 and 1999, who were eligible for vaccination during the public subsidy period, reached approximately 70%, it declined markedly for girls born after 2000; the rate plummeted to below 1% for those born after 2002 [10–13].

The Japanese Ministry of Health, Labour, and Welfare (MHLW, Tokyo, Japan) reinstated the official active recommendation for HPV vaccination in November 2021. In addition, a catch-up HPV vaccination program was launched in April 2022 for women who missed the HPV vaccination opportunity. Nevertheless, without improved health literacy on cervical cancer and the HPV vaccine, young women are less likely to receive vaccination [12, 14–16]. Awareness regarding the risk of cervical cancer should be raised, and people should be educated on the benefits of vaccination to overcome this issue [17–20]. Social media interventions may or may not improve vaccine uptake. Some studies have indicated that increased awareness is not always related to increased vaccine uptake [21–23]. Cooper et al. [24] demonstrated that various digital interventions, including those using social media, have been developed over the past decade to enhance vaccine acceptance and uptake. In low socioeconomic status (SES) families, the campaign reduced uptake by 10%, whereas in low-medium SES families, uptake increased by 6% [25]. HPV promotion campaigns on social media might be a double-edged sword, depending on the target population [25]. Despite the tremendous advantage of implementing HPV vaccination programs for young women, there has been insufficient progress in addressing this issue worldwide, including in Japan.

Therefore, we used different health information delivery methods to enhance the understanding of cervical cancer and HPV vaccination. To the best of our knowledge, this is the first randomized trial to examine whether knowledge and HPV vaccination rates increased after presenting HPV information.

## Materials and methods

### Study design

This study, conducted in our laboratory at Teikyo University between May 1, 2019, and March 31, 2024, implemented a cluster, randomized, parallel cluster randomized group trial involving three groups: Group 1, no intervention (control); Group 2, print-based educational intervention; and Group 3, social networking service (SNS)-based educational intervention. These groups were compared to identify potential educational effects (S1 Fig). The study employed cluster-level randomization; however, due to factors such as the small sample size and the non-randomized recruitment process, the design does not fully conform to the characteristics of a traditional randomized controlled trial (RCT). Despite these limitations, the design was chosen to assess the effectiveness of the interventions while considering ethical constraints and practical challenges.

## Study settings and participants

This study enrolled students from 17 private universities (S2 Fig). Students were assured that study participation was voluntary and would not interfere with their academic activities. We included female students aged 18–26 years who could access and use print-based or SNS-based educational programs and complete the follow-up questionnaire comprising 20 questions and the Communicative and Critical Health Literacy (CCHL) scale. We excluded students who experienced mental and physical challenges during the study. According to the research protocol, students with mental or physical challenges were to be excluded. However, in practice, the study only included students who volunteered to participate, and no students with psychological or physical difficulties were actually enrolled. This exclusion criterion did not affect the study's outcome, as no participants in the study were found to meet this criterion. The universities were selected based on logistical considerations and feasibility. While the focus on private universities facilitated recruitment and ensured voluntary participation, this choice may limit the generalizability of the findings. Due to funding constraints and logistical challenges, the recruitment pool was not expanded beyond these institutions.

## Estimation of the participant number required

The proportion of high health literacy in the competing hypotheses was P1 (intervention group) = 0.60 and P2 (control group) = 0.30, indicating a significant difference between the groups. The within-class correlation, mean cluster size, two-sided significance level, and power were 0.100, 200, 0.05, and 0.80, respectively.

## Intervention

Medical information and educational tools on cervical cancer and HPV vaccination were developed by the principal investigator and distributed to the female students every 6 months. Students were randomly assigned to three arms: arm 1, no intervention (control); arm 2, information was mailed in an envelope containing print-based educational material; and arm 3, information was distributed through electronic materials on websites using social networking sites (LINE, Facebook, and Twitter) (S1 Fig). Students in all three arms were followed up for 15 months, during which, interventions in arms 2 and 3 were conducted three times at 6-month intervals. At the time of recruitment, it was explained to participants that if they were assigned to arm 1, they would only be registered to participate and would not be provided with any information.

## Allocation method

Allocators prepared an allocation table using stratified block randomization with medical and non-medical institutions (Teikyo University and Teikyo Institute of Advanced Nursing Studies, Tokyo, Japan, along with Teikyo Heisei University, Tokyo, Japan) as two stratification factors. The study was an open-label randomized trial, indicating that both allocators and participants were aware of group assignments.

## Data collection

A questionnaire was sent to individuals randomly allocated across the no intervention, print-based educational intervention, and SNS-based educational intervention groups at baseline. All groups received the subsequent survey questionnaires via email, including a questionnaire regarding affiliation, age, diet, HPV vaccination history, the presence of a family healthcare provider, smoking habits, voluntary exercise, physician visits, and routine and HPV

vaccinations (S1 Method), as well as a questionnaire regarding knowledge about cervical cancer (S1 Table). In total, five surveys were administered during the study period.

## Measurements

For the questionnaire about knowledge regarding cervical cancer (S1 Table), "0" was set for the answer "I know" and "1" for "I did not know," and the total score of the 20 items was used as the objective number. Health literacy was measured using the CCHL scale (S2 Table). This scale measures health literacy beyond the functional level, focusing on the ability to access, understand, and use health information; its reliability and validity have been previously confirmed [26]. Participants responded to five questions on the CCHL scale, with scores ranging from 1 (very easy) to 5 (very difficult), focusing on HPV vaccination and cervical cancer screening. The within-individual health literacy score was calculated as the mean of the scores for these five questions. A score of ≤3 was considered 'highly health literate' because it represented participants who had significant knowledge within both groups. The number of participants classified as "high" and "low" will be approximately half at the time of the first survey. As the total number of participants was smaller than expected, we considered it would be difficult to find significant between-group differences if the participant number was skewed toward one group or the other. On the CCHL scale, the average of the 1–5 rating scale was used as another objective rating figure.

## Outcomes

Outcome measures pertained to the cluster level and individual participant level.

**Primary outcome.** The primary outcome was increased knowledge regarding susceptibility to cervical cancer, disease severity, and the benefits of vaccination, as assessed using the reliable and valid health belief model [26]. The primary endpoint was the proportion of participants with high health literacy scores immediately after the delivery of the third educational session at 12 months.

**Secondary outcome.** We investigated how participant backgrounds and family environments affected the sum of their questionnaire responses regarding knowledge of cervical cancer and the mean value of the CCHL scale score.

## Efforts to avoid possible sources of bias

Bias may have been introduced due to the self-selection of survey participants. Thus, we asked all students to enter this research voluntarily through a public post in the university or other notices, such that every student had access to the information. To avoid biases related to student environments, the allocation of participants to the three groups was conducted using an allocation table.

## Main analysis

A mixed-effect logistic model was used for data analysis, with the dichotomous variable indicating high or low health literacy serving as the outcome variable. The model calculated the odds ratio to determine the effect of the intervention on high health literacy, considering various factors, including the experimental group, institution, and medical and non-medical background as population effects, and cluster as the variable effect. An odds ratio that was significantly higher than 1 indicated that the intervention positively influenced health literacy. Age and prior knowledge were optional adjustment factors that could be incorporated into the model based on the discretion of the statistical analyst and principal investigator. A

significance level of 0.05 was set for all statistical tests. The one-way analysis of variance was performed for continuous variables, and the Chi-square test was performed for categorical variables in the subgroup analysis. For the sub-analysis, we examined the literacy level in each cluster by collecting the total scores of the first 20 questionnaires returned for each cluster in the first survey. In the questionnaires, participants responded with "Yes, I know," or "No, I don't know" to each item, which were scored as 0 or 1, respectively. These scores were summed; a lower total score indicated that the respondents were more knowledgeable about HPV, cervical cancer, and HPV vaccination. It is important to note that, unlike typical scoring systems where higher scores reflect better knowledge, in this study, a lower score reflects higher knowledge. Statistical analyses were performed using JMP Pro version 17.0.0 (SAS Institute Inc., Cary, NC, USA).

## Ethical considerations

This study was approved by the Institutional Review Board of Teikyo University (protocol code 18-195-3; date of approval: March 22, 2019). Although a data-monitoring committee was not deemed necessary for this feasibility study, as we did not anticipate any adverse events, any unintended consequences of the interventions were diligently recorded. All students and their parents provided written informed consent before participation.

## Results

### Participant selection

Overall, 15,400 young female students were targeted and recruited to participate in the study. Consequently, due to student behavioral restrictions at the university, 267 participants met the eligibility criteria and 79 did not; thus, the data of 188 participants were included in the analyses. The invitation to participate was only posted in front of the educational affairs division due to the challenge of making direct contact with students.

### Participant characteristics

Of the 267 participants, 188 completed all questionnaires 3 times (S3 Table). The mean age was 21 (standard deviation 1.67) years. Participant affiliations included Nursing (27.1%), Medicine (23.9%), Pharmaceutical (20.7%), Medical technology (14.9%), Literature (8.0%), Economics (2.7%), Law (1.6%), Foreign Language (0.5%), and Science and Technology (0.5%). Those who had any medical professionals in their family accounted for 38.3%. Non-smokers accounted for 98.8%, and 67.6% were aware of the importance of a balanced diet. Regarding exercise frequency, 60.6% did not exercise, and 15.4%, 19.7%, and 2.66% exercised once a week, two or three times a week, and daily, respectively. A total of 80.3% had not consulted an obstetrician or gynecologist before; 81.9% had routine vaccinations conducted in accordance with Japanese law; 56.9% had not been vaccinated for HPV; and 9.6%, 7.4%, and 25.5% had received an HPV vaccination once, twice, and thrice, respectively.

### Questionnaire

Each participant was randomly assigned to one of three groups. Questionnaires were sent out at enrolment (baseline) in the manner assigned to the groups. The survey was conducted in three rounds, and the results from the 141 participants who completed all three rounds are presented in S4 Table. Responses to Item 11 in the questionnaire indicated a significant improvement in the participants' knowledge. Specifically, participants became more aware that malignant findings in the cervix and procedures including conization could increase the

risk of imminent miscarriage and premature birth. The responses to Item 16 also illustrated a significant improvement in understanding regarding the efficiency of "catch-up vaccination."

However, responses to Items 2 and 8 demonstrated that a lower percentage of participants had knowledge of HPV types, including high-risk HPV, and the mortality rate of cervical cancer in both the first and third surveys than in the others, indicating that students had difficulty finding the exact figures related to cervical cancer (S5 Table). Despite the common belief that early detection of cervical lesions is necessary for treatment, surprisingly, approximately half of the participants were unaware of the perinatal risks associated with surgery involving the uterus.

## CCHL scale

The CCHL scale results are presented in Table 1. There were no inquiries from participants about the questions. In the first and second survey rounds, most students rated the five items on the scale as "easy" or "slightly easy." However, many students responded "very difficult" to Items 3 and 4, which were related to being able to understand and communicate information related to the HPV vaccine and cervical cancer screening to others and being able to determine which information is reliable or not.

## Main outcome

The null hypothesis stated that the proportion of participants with high health literacy in Groups 1, 2, and 3 would be equal. Conversely, the alternative hypothesis posited that the proportion of participants with high health literacy in Groups 2 and 3 would be greater than that in Group 1, which was a one-sided hypothesis. Logistic regression analysis applying a mixed-effects model showed that the odds ratio for "high health literacy" was below 1 for the print-based and control groups. Interestingly, this difference did not reach statistical significance for the groups that seemed more interested in HPV vaccination: participants who had visited a gynecologist, received HPV vaccination, and completed three HPV vaccinations (Fig 1, S4 Table). Intriguingly, participants who were routinely aware of diet importance had an odds ratio of <1, suggesting that they have lower motivation for increasing their knowledge and collecting information on the HPV vaccine and cervical cancer (Fig 1).

## Subgroup analyses

We found significant differences in the total scores of the questionnaire depending on whether participants were students at medical facilities and whether they had received HPV vaccinations before the study (Fig 2a–2d). However, no significant differences were found between the average CCHL scale scores obtained in the first survey between subgroups defined by HPV vaccination status (Fig 2e and 2f).

We then examined participants' knowledge improvement during the study period using the average of the differences in total scores between the first and third surveys, based on the first 20-item questionnaire completed about knowledge regarding cervical cancer, received in each cluster. No significant differences were found among the three education groups, whereas students in a non-medical faculty tended to show greater improvements compared with those in a medical faculty (S3 Fig).

We then compared each group based on the average of the differences in the CCHL scale scores between the first and third surveys. We found no significant difference among the three groups. Interestingly, in the print-based group, no improvement was observed; rather, a slight regression in literacy was noted (Fig 3a and 3b).

**Table 1. Results of the Communicative and Critical Health Literacy Scale in the first, second, and third survey rounds.**

| | | Degree of difficulty *N (%) (N = 141) | | | | |
|---|---|---|---|---|---|---|
| | | Very easy | Slightly easy | Intermediate | Slightly difficult | Very difficult |
| 1) I can collect information related to the HPV vaccine and cervical cancer screening from various sources, such as newspapers, books, television, and the Internet. | 1st | | | | | |
| | | 45 (31.9) | 61 (43.3) | 18 (12.8) | 14 (9.9) | 2 (1.4) |
| | 2nd | | | | | |
| | | 35 (24.8) | 58 (41.1) | 23 (16.3) | 23 (16.3) | 0 (0) |
| | 3rd | | | | | |
| | | 29 (20.6) | 71 (50.4) | 28 (19.9) | 13 (9.2) | 0 (0) |
| 2) I can extract the information you are looking for from a large selection of information related to the HPV vaccine and cervical cancer screening. | 1st | | | | | |
| | | 11 (7.8) | 52 (36.9) | 36 (25.5) | 35 (24.8) | 6 (4.3) |
| | 2nd | | | | | |
| | | 16 (11.3) | 50 (35.5) | 28 (19.9) | 39 (27.7) | 6 (4.3) |
| | 3rd | | | | | |
| | | 13 (9.2) | 55 (35.5) | 35 (24.8) | 34 (24.1) | 4 (2.8) |
| 3) I can understand and communicate the obtained information related to the HPV vaccine and cervical cancer screening to others. | 1st | | | | | |
| | | 8 (5.7) | 33 (23.4) | 35 (24.8) | 44 (31.2) | 20 (14.2) |
| | 2nd | | | | | |
| | | 12 (8.5) | 29 (20.6) | 48 (34.0) | 32 (22.7) | 16 (11.3) |
| | 3rd | | | | | |
| | | 9 (6.4) | 36 (25.5) | 38 (27.0) | 46 (32.6) | 11 (7.8) |
| 4) I can judge the credibility of the information related to the HPV vaccine and cervical cancer screening. | 1st | | | | | |
| | | 6 (4.3) | 23 (16.3) | 37 (26.2) | 55 (39.0) | 19 (13.5) |
| | 2nd | | | | | |
| | | 9 (6.4) | 30 (21.3) | 38 (27.0) | 47 (33.3) | 15 (10.6) |
| | 3rd | | | | | |
| | | 9 (6.4) | 40 (28.4) | 36 (25.5) | 46 (32.6) | 11 (7.8) |
| 5) I can make decisions about plans and actions for improving my health based on information related to the HPV vaccine and cervical cancer screening. | 1st | | | | | |
| | | 12 (8.5) | 37 (26.2) | 44 (31.2) | 36 (25.5) | 11 (7.8) |
| | 2nd | | | | | |
| | | 13 (9.2) | 52 (36.9) | 36 (25.5) | 30 (21.3) | 7 (5.0) |
| | 3rd | | | | | |
| | | 9 (6.4) | 55 (39.0) | 52 (36.9) | 24 (17.0) | 0 (0) |

HPV, human papillomavirus

*The number of participants who responded to these items.

Additionally, the group comparisons for whether participants had medical experts as family members yielded similar results. In the group of participants with medical professionals as family members, a slight regression was observed in the difference in CCHL scale scores between the first and third surveys (Fig 3c–3e).

## Discussion

In this study, we surveyed the effect of an education intervention on Japanese female students' knowledge regarding HPV vaccination and cervical cancer for the first time in Japan. We

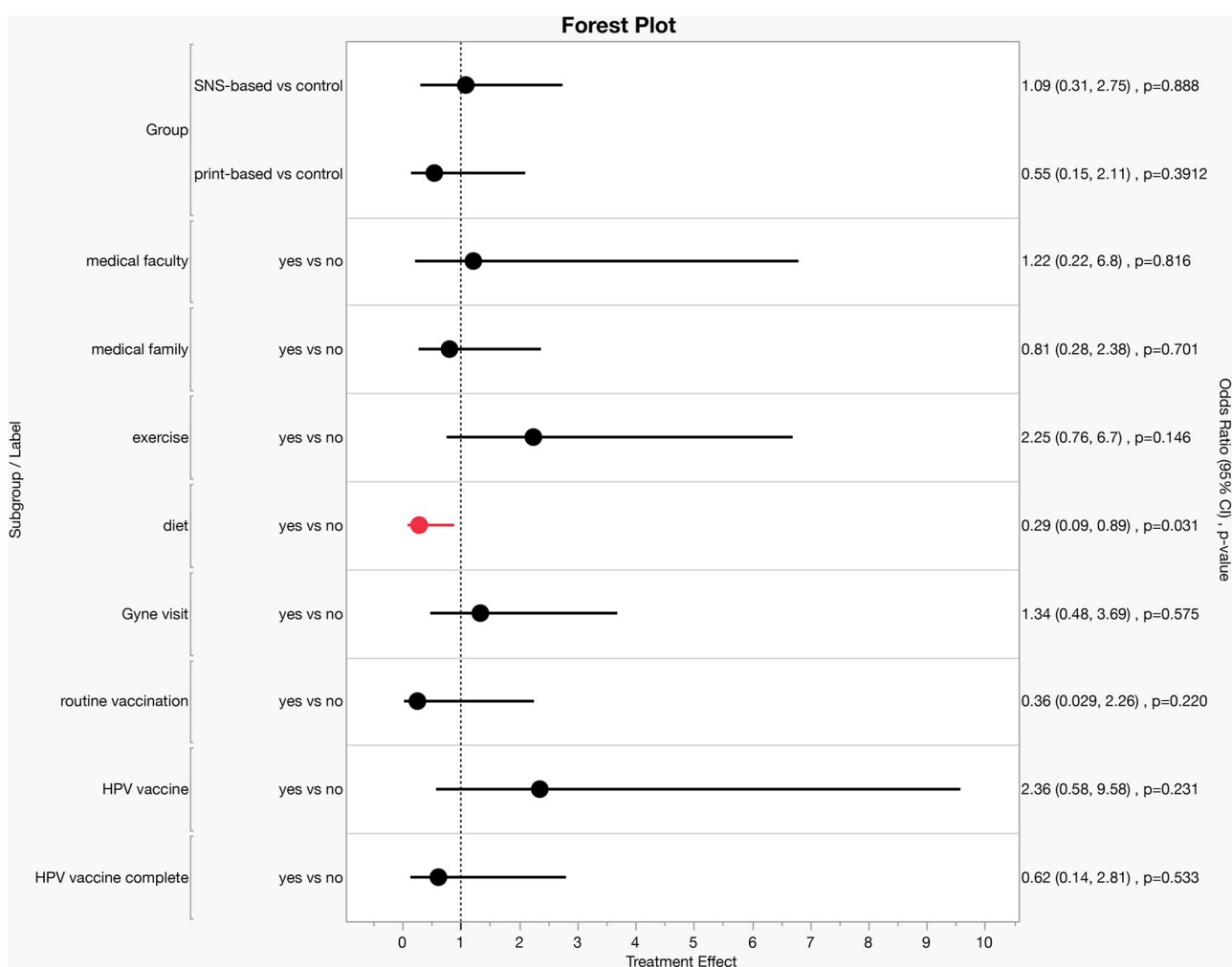

**Fig 1. Odds ratios for "high health literacy" in the third survey, adjusted for CCHL scale scores.** Odds ratio (OR) for being "highly health literate" in the third survey. Logistic regression analysis was used to estimate the adjusted ORs and 95% confidence intervals (CIs) for Communicative and Critical Health Literacy (CCHL) scale scores (response variable: 1 = CCHL "high" in the 3rd survey, 0 = "low").

found that participant knowledge and health literacy improved regardless of whether correct education about the HPV vaccine and cervical cancer was presented through print material or SNS. The analysis comparing the responses to the first and third surveys in the three groups showed that health literacy increased substantially, even in the control group that received no specific educational intervention. The improvement in health literacy in the control group indicated that the realization of "not knowing" was an important factor, that can drive students to investigate, think, and learn further information on their own accord. Conversely, the analysis comparing the differences in CCHL scale scores showed no improvement in the print-based group but rather a slight regression over time. As presented in Fig 3a, health literacy in the print group tended to be slightly higher from the start. However, their improvement in scores was less than that in the other groups; thus, print is not read repeatedly or does not stimulate intellectual interest, even when it is handed out. Although the insufficient numbers hamper a definite conclusion, the initial health literacy was higher in the print-based group than in the other groups, which suggests that already health-literate participants were less likely to obtain further information. However, further research on this topic is required.

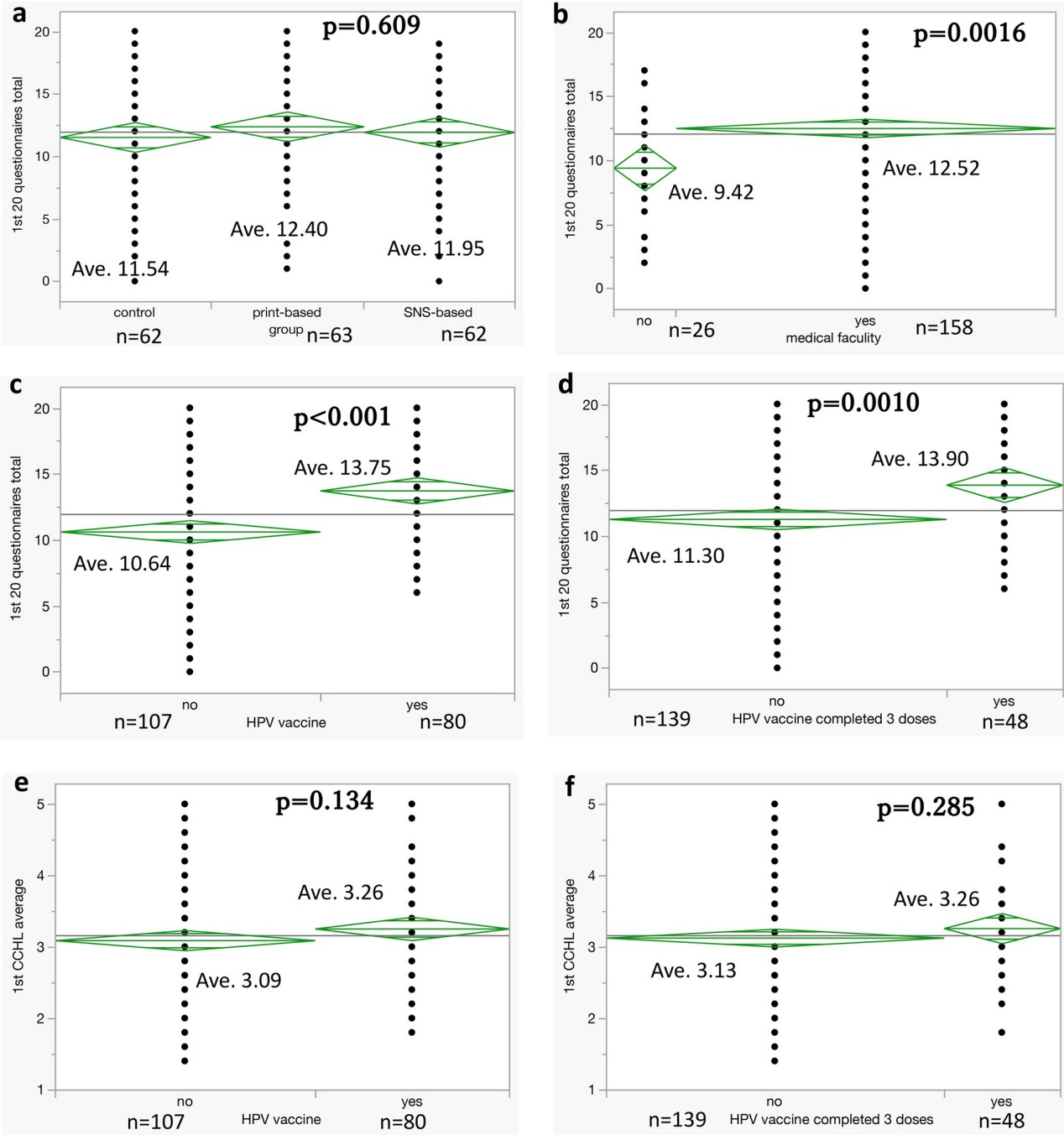

**Fig 2. Subgroup analyses of questionnaire total scores and CCHL scale scores.** a–d: There are significant differences in the total scores of the questionnaire depending on whether participants were students at medical facilities and whether they had received HPV vaccinations before the study. e, f: No significant differences were found between the average Communicative and Critical Health Literacy (CCHL) scale scores obtained in the first survey between subgroups defined by HPV vaccination. HPV complete group: those who had already completed three doses of the HPV vaccine at the start of the study.

A previous study aimed at determining the factors that influence satisfaction with decision-making concerning HPV vaccination among female university students in Japan concluded that being vaccinated against HPV, having higher knowledge scores, and having lower awareness regarding the risk of sexually transmitted infections were significant influencing factors

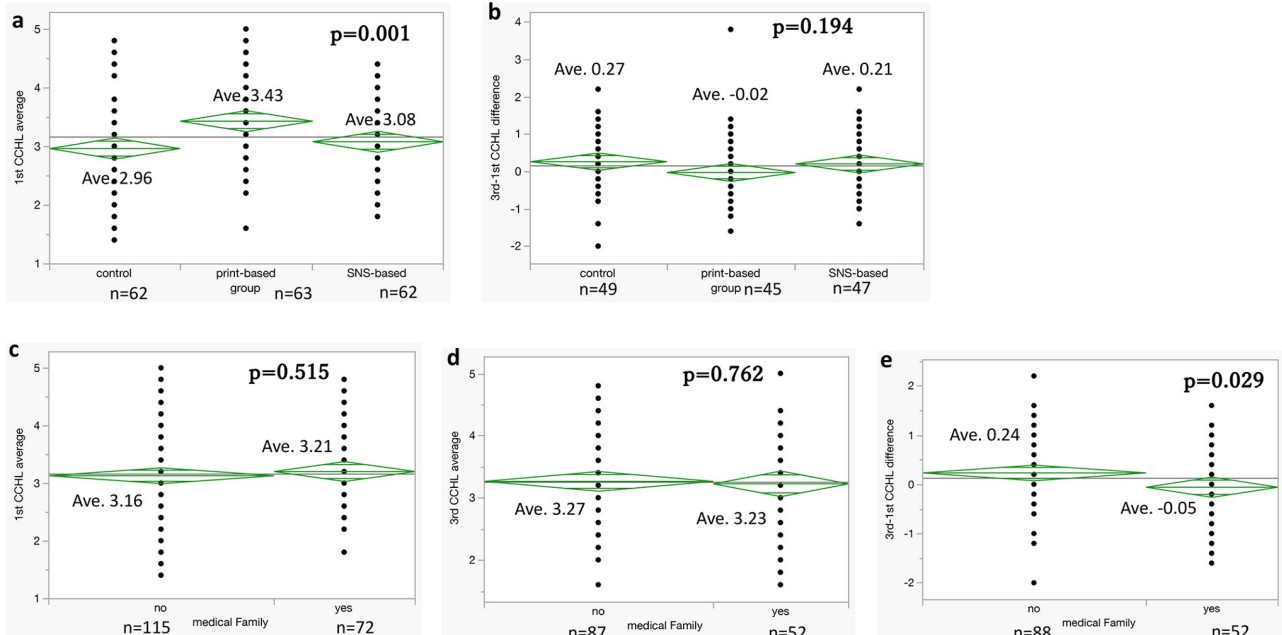

**Fig 3. Comparison of average differences in CCHL scale scores.** a, b: Comparison of groups based on the average of the differences in the Communicative and Critical Health Literacy (CCHL) scale scores between the first and third surveys. There are no significant differences among the three groups. Interestingly, in the print-based group, no improvement can be observed; rather, a slight regression in literacy can be noted. c–e: Comparison of average CCHL scores in the first and third surveys, and the difference between the first and third surveys, in groups defined by the presence of family members in the medical profession. In the group of participants with medical professionals as family members, a slight regression can be observed in the difference in CCHL scale scores between the first and third surveys. c: First questionnaire. d: Third questionnaire. e: Difference between the first and third questionnaires. The difference is more pronounced in the negative, with a higher improvement observed when there is no family member in the medical profession.

[27]. In Japan, the need to make appropriate decisions and behavioral choices according to scientific thinking and the correct judgment of health-related issues is discussed in the school education curricula [28]. Nevertheless, considering that the recommended target population for HPV vaccination is students from the sixth grade of elementary school to the first year of high school, the students, as well as their parents, should be targeted for intervention. Therefore, the family environment is an important factor in health literacy.

In our study, subgroup analyses of the groups with and without a family member in the medical profession showed that the mean CCHL scale score tended to be slightly higher for the first survey in the subgroup with a family member in the medical profession (Fig 3c). However, for the third survey, the mean CCHL scale scores were reversed, with greater improvement in the subgroup without a family member in the medical profession (Fig 3e). This indicates that, in modern Japan, people can absorb the correct information if they are alerted by their surroundings and are willing to learn independently.

A study of Swiss university students suggested that primary care has a high potential for increasing HPV vaccination coverage rates [29]. Unlike in countries in Europe and the United States, Japanese citizens, particularly the younger generation, do not have a "family doctor." Therefore, the solution for vaccination uptake lies in the educational environment in the family, local community, and school. Social media platforms should be arranged so that young students and their parents can obtain appropriate and sufficient information, regardless of their situation.

In their investigation of strategies to debunk vaccine untruths, paradoxical effects were reported by Betsch and Sachse [30]. However, subsequent investigations failed to provide any

scientific or epidemiological evidence supporting a causal relationship between the reported symptoms, such as pain and motor dysfunction, and HPV vaccination [31]. Consequently, the Ministry of Health, Labour and Welfare (MHLW) Adverse Effects Review Committee confirmed that these symptoms were functional physical symptoms [32, 33]. A nationwide epidemiological survey conducted by the Sobue Group of the MHLW also reported that similar symptoms to those reported post-vaccination were present among individuals without a history of HPV vaccination [34]. In a questionnaire survey of women born between 1994 and 2000 in Nagoya, no significant difference in the age-adjusted incidence of 24 symptoms was found between vaccinated and unvaccinated women, providing no evidence of a causal relationship between the symptoms and HPV vaccination [35]. The Japanese government's decision to stop the active recommendation of HPV vaccination spanned 8 years, from June 2013 to November 2021 [36]. Meanwhile, Australia, Sweden, and Denmark published the results of the effects and safety of HPV vaccination, reporting that a substantial reduction was anticipated in the risk of invasive cervical cancer among vaccinated women [37–40]. Knowledge and awareness regarding cervical cancer and HPV vaccines in various countries have been analyzed, and the research in Japan remains limited [41–52]. The results of this study are expected to enhance knowledge and awareness regarding cervical cancer among young women in Japan.

## Limitations

This study is limited by the relatively small sample size, which was partly due to the ethical care taken in the recruitment methods. Utmost care was taken to solicit voluntary participation without coercion or harassment against students. Recruitment was also affected by the COVID-19 pandemic when many students were not attending class. Thus, the number of participants was much lower than expected, resulting in a lack of statistically significant differences due to the low study power. Nevertheless, this study provides valuable information. Additionally, the study participants were recruited from young female university students who were asked to participate voluntarily. Therefore, the target population was limited to university students who were interested in this study; inevitably, this involved bias in terms of family environment, parental income, and access to information sources. Thus, the results may not be generalizable to all Japanese women. This study was a physician-initiated research project funded by an external agency. It is important to note that the funding agency had no influence over the study design, participant recruitment, data collection, or analysis. All participants were recruited voluntarily and were not selected based on the funding agency's preferences. The funding agency had no impact on the study results or the interpretation of the findings. This transparency regarding the role of the funding agency reinforces the credibility of the study's conclusions, ensuring that the study's design and outcomes were independent of any external influence.

## Conclusion

Our analysis indicated that participants' knowledge and health literacy on cervical cancer and the HPV vaccine improved regardless of whether education was delivered by a print-based or SNS-based intervention; participants acquired proper knowledge and awareness regarding the importance of the disease and its prevention. These findings highlight the effectiveness of both traditional and digital educational methods in enhancing public health literacy. Moving forward, such interventions can be integrated into broader public health campaigns to improve cervical cancer prevention, especially in populations with limited access to healthcare education.

## Supporting information

**S1 Fig. Study procedure.**
(TIF)

**S2 Fig. Study flowchart.**
(TIF)

**S3 Fig. Comparison of average differences in total scores between the first and third surveys, based on the first 20-item questionnaire completed about knowledge regarding cervical cancer received in each cluster.** No significant differences were found among the three education groups, whereas non-medical students faculty tend to show greater improvements than did those in a medical faculty.
(TIF)

**S1 Method. Questions about your current life.**
(PDF)

**S1 Table. Questionnaire about knowledge of cervical cancer and the human papillomavirus vaccine.**
(PDF)

**S2 Table. Communicative and Critical Health Literacy Scale (CCHL).**
(PDF)

**S3 Table. Baseline characteristics of the female students who participated in the study from three universities in Japan, 2019–2023 (N = 188: Number of participants).**
(PDF)

**S4 Table. Factors related to "high" health literacy scores in the third survey.**
(PDF)

**S5 Table. Distribution of knowledge regarding cervical cancer items and rate of the answer "I know" in the first, second, and third rounds (N = 141).**
(PDF)

## Acknowledgments

The authors are grateful for helpful discussions with former colleagues Dr. Shiho Fukui, Dr. Yoshiko Kawata (Department of Obstetrics and Gynecology, University of Tokyo), and Prof. Kyoko Nomura (Department of Public Health, Akita University Graduate School of Medicine). We thank our students and colleagues at Teikyo University for their critical comments. For recruiting students to participate in the study, the teaching departments of Teikyo University, Teikyo Heisei University, and Teikyo Institute of Advanced Nursing provided tremendous support by displaying recruitment posters.

## Author Contributions

**Conceptualization:** Yuko Takahashi, Kazunori Nagasaka.

**Data curation:** Yuko Takahashi, Hirono Ishikawa, Kazunori Nagasaka.

**Formal analysis:** Yuko Takahashi, Yuko Miyagawa, Kazunori Nagasaka.

**Funding acquisition:** Kazunori Nagasaka.

**Investigation:** Yuko Takahashi, Yukifumi Sasamori, Risa Higuchi, Asumi Kaku, Tomoo Kumagai, Saya Watanabe, Miki Nishizawa, Kazuki Takasaki, Haruka Nishida, Takayuki Ichinose, Mana Hirano, Yuko Miyagawa, Haruko Hiraike, Koichiro Kido, Kazunori Nagasaka.

**Methodology:** Yuko Takahashi, Hirono Ishikawa, Kazunori Nagasaka.

**Project administration:** Kazunori Nagasaka.

**Resources:** Yuko Takahashi, Yukifumi Sasamori, Yuko Miyagawa, Haruko Hiraike, Kazunori Nagasaka.

**Software:** Yuko Takahashi, Yuko Miyagawa, Kazunori Nagasaka.

**Supervision:** Hirono Ishikawa, Kazunori Nagasaka.

**Validation:** Yuko Takahashi, Yuko Miyagawa, Hirono Ishikawa, Kazunori Nagasaka.

**Visualization:** Kazunori Nagasaka.

**Writing – original draft:** Yuko Takahashi.

**Writing – review & editing:** Kazunori Nagasaka.

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
