## [Decision Letter · Decision Letter 0]

15 Oct 2024

PONE-D-24-35490Effects of different educational interventions on cervical cancer knowledge and human papilloma virus vaccination uptake among young women in Japan: preliminary results of a cluster randomized controlled trialPLOS ONE

Dear Dr. Nagasaka,

Thank you for submitting your manuscript to PLOS ONE. After careful consideration, we feel that it has merit but does not fully meet PLOS ONE’s publication criteria as it currently stands. Therefore, we invite you to submit a revised version of the manuscript that addresses the points raised during the review process.

We look forward to receiving your revised manuscript.

Kind regards,

Lucy W. Kivuti-Bitok, Ph.D. MHSM,BScN

Academic Editor

PLOS ONE

Journal Requirements: When submitting your revision, we need you to address these additional requirements. 1. Please ensure that your manuscript meets PLOS ONE's style requirements, including those for file naming. The PLOS ONE style templates can be found at https://journals.plos.org/plosone/s/file?id=wjVg/PLOSOne_formatting_sample_main_body.pdf and https://journals.plos.org/plosone/s/file?id=ba62/PLOSOne_formatting_sample_title_authors_affiliations.pdf 2. Thank you for stating the following financial disclosure: "This research was funded in part by the Investigator-Initiated Studies Program of Merck Sharp & Dohme Corp. (Kenilworth, NJ, USA) and MSD K.K. (grant number 58246). The opinions expressed in this study are those of the authors and do not necessarily represent those of Merck Sharp & Dohme Corp. or MSD K.K." Please state what role the funders took in the study.  If the funders had no role, please state: ""The funders had no role in study design, data collection and analysis, decision to publish, or preparation of the manuscript."" If this statement is not correct you must amend it as needed. Please include this amended Role of Funder statement in your cover letter; we will change the online submission form on your behalf. 3. Thank you for stating in your Funding Statement: "This research was funded in part by the Investigator-Initiated Studies Program of Merck Sharp & Dohme Corp. (Kenilworth, NJ, USA) and MSD K.K. (grant number 58246). The opinions expressed in this study are those of the authors and do not necessarily represent those of Merck Sharp & Dohme Corp. or MSD K.K." Please provide an amended statement that declares *all* the funding or sources of support (whether external or internal to your organization) received during this study, as detailed online in our guide for authors at http://journals.plos.org/plosone/s/submit-now.  Please also include the statement “There was no additional external funding received for this study.” in your updated Funding Statement. Please include your amended Funding Statement within your cover letter. We will change the online submission form on your behalf. 4. Thank you for stating the following in the Competing Interests section: "I have read the journal's policy and the authors of this manuscript have the following competing interests:This research was funded in part by the Investigator-Initiated Studies Program of Merck Sharp & Dohme Corp. (Kenilworth, NJ, USA) and MSD K.K. (grant number 58246). The opinions expressed in this study are those of the authors and do not necessarily represent those of Merck Sharp & Dohme Corp. or MSD K.K." Please confirm that this does not alter your adherence to all PLOS ONE policies on sharing data and materials, by including the following statement: ""This does not alter our adherence to  PLOS ONE policies on sharing data and materials.” (as detailed online in our guide for authors http://journals.plos.org/plosone/s/competing-interests).  If there are restrictions on sharing of data and/or materials, please state these. Please note that we cannot proceed with consideration of your article until this information has been declared.  Please include your updated Competing Interests statement in your cover letter; we will change the online submission form on your behalf. 5. Please review your reference list to ensure that it is complete and correct. If you have cited papers that have been retracted, please include the rationale for doing so in the manuscript text, or remove these references and replace them with relevant current references. Any changes to the reference list should be mentioned in the rebuttal letter that accompanies your revised manuscript. If you need to cite a retracted article, indicate the article’s retracted status in the References list and also include a citation and full reference for the retraction notice.

Reviewers' comments:

Reviewer's Responses to Questions

**Comments to the Author**

1. Is the manuscript technically sound, and do the data support the conclusions?

Reviewer #1: Yes

Reviewer #2: Yes

2. Has the statistical analysis been performed appropriately and rigorously? 

Reviewer #1: Yes

Reviewer #2: Yes

3. Have the authors made all data underlying the findings in their manuscript fully available?

Reviewer #1: Yes

Reviewer #2: Yes

4. Is the manuscript presented in an intelligible fashion and written in standard English?

Reviewer #1: Yes

Reviewer #2: Yes

5. Review Comments to the Author

Reviewer #1: I recommend this manuscript for publication with minor revisions and some significant concerns.

The sample size is a significant concern. The authors mention the small number of participants as a limitation, but the reasons provided are not sufficiently compelling. While I recognize the challenges posed by the COVID-19 pandemic, more effort could have been made to increase the number of participants. For instance, broadening the recruitment pool beyond private university students could have provided more robust data and increased generalizability. Additionally, the rationale for selecting only private universities is not clearly articulated. This raises questions about whether the findings of this study can be referred to as representative of the population. It also raises concerns about potential selection bias. Expanding on why these institutions were chosen and how this decision may affect the study’s applicability to other populations would strengthen the argument.

In the “Study settings and participants” section, it is noted that students facing mental or physical challenges were excluded, but there is no detailed explanation of what specific challenges were encountered or how their exclusion may have influenced the findings. The study would benefit from further elaboration on this point, including any possible connections between the excluded students’ challenges and the study’s outcomes. Clarifying this could help readers understand whether the exclusion of these participants had any significant impact on the study’s results.

Regarding the classification of the study as a randomized controlled trial (RCT), I have reservations. The study design, combined with the small sample size and the recruitment strategy, makes it difficult to categorize this work as an RCT. The control mechanisms seem more aligned with a case-control study, given the observational nature of the data and the lack of a fully randomized, robust sample size. The authors should reconsider framing the study as an RCT or, at the very least, revisit their conclusions to reflect the study’s limitations in this regard.

Lastly, while the authors declare that the funding agency did not influence the study design, data collection, or analysis, I would appreciate more discussion regarding the role of the funding body in the study’s limitations section. A transparent reflection on how the presence of a funder might have influenced certain study decisions would provide additional credibility to the work.

Overall, the paper has merit, and with the suggested revisions, it has the potential to strengthen its contribution to the field. I encourage the authors to consider these points and revise the manuscript accordingly.

Reviewer #2: Reviewer comments for the Manuscript Number PONE-D-24-35490

“Effects of different educational interventions on cervical cancer knowledge and human papilloma virus vaccination uptake among young women in Japan: preliminary results of a utcluster randomized controlled trial”

Kazunori Nagasaka et al

The manuscript is well written but a few areas need clarification . This way it will make it easier for the reader and enhance clarity and flow in the article.The attached file contains the areas that need attention

6. PLOS authors have the option to publish the peer review history of their article (what does this mean?). If published, this will include your full peer review and any attached files.

Reviewer #1: **Yes: **Dr. Shamim Ahmed

Reviewer #2: No

---

## [Author Response · Author response to Decision Letter 0]

23 Nov 2024

Dear Reviewers,

We wish to re-submit the manuscript titled “Effects of different educational interventions on cervical cancer knowledge and human papillomavirus vaccination uptake among young women in Japan: preliminary results of a cluster randomized controlled trial.” The manuscript has been rechecked and the necessary changes have been made in accordance with the reviewers’ suggestions. The responses to all comments have been prepared and attached herewith. 

Thank you for your consideration. I look forward to hearing from you.

Sincerely,

Kazunori Nagasaka MD, PhD

Reviewer #1: I recommend this manuscript for publication with minor revisions and some significant concerns.

The sample size is a significant concern. The authors mention the small number of participants as a limitation, but the reasons provided are not sufficiently compelling. While I recognize the challenges posed by the COVID-19 pandemic, more effort could have been made to increase the number of participants. For instance, broadening the recruitment pool beyond private university students could have provided more robust data and increased generalizability. Additionally, the rationale for selecting only private universities is not clearly articulated. This raises questions about whether the findings of this study can be referred to as representative of the population. It also raises concerns about potential selection bias. Expanding on why these institutions were chosen and how this decision may affect the study’s applicability to other populations would strengthen the argument.

→We appreciate the reviewer’s insightful feedback regarding the sample size and recruitment strategy. We acknowledge that the relatively small sample size could be seen as a limitation. While the COVID-19 pandemic presented significant challenges to participant recruitment and study expansion, we agree that more effort could have been made to increase the number of participants. Originally, we intended to broaden the recruitment pool beyond private universities and planned to collaborate with additional universities and educational institutions, including junior high schools and high schools in Tokyo. However, due to funding constraints and difficulties in securing cooperation from other institutions, we were unable to execute this broader recruitment strategy. Furthermore, public awareness of HPV vaccination was limited at the time due to government guidelines, which impacted participant engagement. Regarding the selection of private universities, our decision was influenced by several factors. First, the study aimed to ensure voluntary participation, particularly because students are a vulnerable population, and the Ethics Committee stressed the importance of preventing any undue influence from academic obligations. Recruiting from private universities helped to ensure that participation was voluntary and did not inadvertently pressure students due to academic requirements. Additionally, we considered the feasibility of recruitment within the limited time frame and the available resources. We recognize that this selection may limit the generalizability of our findings to the broader population, and we have added a more thorough explanation of these limitations in the manuscript. We agree that future studies should aim to include a more diverse and representative sample to strengthen the applicability of the findings to other populations. On lines 109-114, we have added the following statement: “The universities were selected based on logistical considerations and feasibility. While the focus on private universities facilitated recruitment and ensured voluntary participation, this choice may limit the generalizability of the findings. Due to funding constraints and logistical challenges, the recruitment pool was not expanded beyond these institutions.” 

In the “Study settings and participants” section, it is noted that students facing mental or physical challenges were excluded, but there is no detailed explanation of what specific challenges were encountered or how their exclusion may have influenced the findings. The study would benefit from further elaboration on this point, including any possible connections between the excluded students’ challenges and the study’s outcomes. Clarifying this could help readers understand whether the exclusion of these participants had any significant impact on the study’s results.

→Thank you for highlighting this point. The research protocol specifies that students with mental or physical challenges were to be excluded. However, in practice, the study only included students who volunteered to participate, and no students with psychological or physical difficulties were actually enrolled. We have clarified this in the text to ensure accurate understanding on the line 105-108: “According to the research protocol, students with mental or physical challenges were to be excluded. However, in practice, the study only included students who volunteered to participate, and no students with psychological or physical difficulties were actually enrolled. This exclusion criterion did not affect the study’s outcome, as no participants in the study were found to meet this criterion.”

Regarding the classification of the study as a randomized controlled trial (RCT), I have reservations. The study design, combined with the small sample size and the recruitment strategy, makes it difficult to categorize this work as an RCT. The control mechanisms seem more aligned with a case-control study, given the observational nature of the data and the lack of a fully randomized, robust sample size. The authors should reconsider framing the study as an RCT or, at the very least, revisit their conclusions to reflect the study’s limitations in this regard.

→Thank you for this valuable insight. As the reviewer noted, this study’s design has inherent limitations that make it challenging to classify as an RCT. We have updated the manuscript to include these limitations in the "Study design" section and have carefully reconsidered the framing of the study’s design as below on line 91-96: “The study employed cluster-level randomization; however, due to factors such as the small sample size and the non-randomized recruitment process, the design does not fully conform to the characteristics of a traditional randomized controlled trial (RCT). Despite these limitations, the design was chosen to assess the effectiveness of the interventions while considering ethical constraints and practical challenges.”

Lastly, while the authors declare that the funding agency did not influence the study design, data collection, or analysis, I would appreciate more discussion regarding the role of the funding body in the study’s limitations section. A transparent reflection on how the presence of a funder might have influenced certain study decisions would provide additional credibility to the work.

Overall, the paper has merit, and with the suggested revisions, it has the potential to strengthen its contribution to the field. I encourage the authors to consider these points and revise the manuscript accordingly.

→We are very grateful for this suggestion. In the "Limitations" section, we have reiterated that this was a physician-initiated study funded by the agency without influence over the design, recruitment, or analysis on line 453-460: “This study was a physician-initiated research project funded by an external agency. It is important to note that the funding agency had no influence over the study design, participant recruitment, data collection, or analysis. All participants were recruited voluntarily and were not selected based on the funding agency’s preferences. The funding agency had no impact on the study results or the interpretation of the findings. This transparency regarding the role of the funding agency reinforces the credibility of the study's conclusions, ensuring that the study's design and outcomes were independent of any external influence.”. All participants were voluntary and not selected based on funder preferences, and the funding agency had no impact on the results. We believe this additional clarification strengthens the transparency of our study.

Reviewer #2: Reviewer comments for the Manuscript Number PONE-D-24-35490

“Effects of different educational interventions on cervical cancer knowledge and human papilloma virus vaccination uptake among young women in Japan: preliminary results of a utcluster randomized controlled trial”

Kazunori Nagasaka et al.

The manuscript is well written but a few areas need clarification . This way it will make it easier for the reader and enhance clarity and flow in the article. The attached file contains the areas that need attention

Reviewer comments for the Manuscript Number PONE-D-24-35490

“Effects of different educational interventions on cervical cancer knowledge and human papilloma virus vaccination uptake among young women in Japan: preliminary results of a utcluster randomized controlled trial”

 Kazunori Nagasaka et al.

Abstract

The abstract looks good and gives an overview of cervical cancer in Japan it could improve by providing more context about the adverse reactions that led to the suspension of the HPV vaccine could enhance understanding of vaccine hesitancy.

→Thank you for this helpful feedback. We have revised the abstract to include details on the adverse reactions that contributed to the suspension of the HPV vaccination, addressing the issue of vaccine hesitancy.

Introduction

This section is good. It can be improved by providing specific examples of how health literacy can be improved. Other areas the author needs to check are for instance;

Line 70 Cooper et al. [24]... the reference style is not consistent with the other references. Ensure that references are consistent and clearly formatted to improve readability of the article.

→We sincerely appreciate the reviewer’s suggestions. We have added specific examples of strategies to improve health literacy. We have also corrected the formatting inconsistency in Line 71 and will ensure that all references are consistently and clearly formatted throughout the article.

Materials and methods

This section looks good but the authors will need to check and make correction on the following.

Line 86 a cluster, randomized, parallel-group trial - This is a bit confusing. did you mean parallel cluster randomized group trial?

→ Thank you for your suggestion. We have modified the wording to "parallel cluster randomized group trial" on line 87 to improve clarity.

Line 121 Clusters were defined as a faculty or department and an allocation. This statement is not clear, is there some information missing?

→ Thank you for pointing this out. We did not conduct a cluster analysis in this study; thus, we have removed this sentence for accuracy.

The authors indicate that Analysis of variance with Scheffé test was performed for continuous variables, and the Chi-square test ….. Was the Scheffé technique utilised in the analysis? If they did were the results presented in the writeud?

→Thank you for highlighting this. We performed a one-way analysis of variance for continuous variables and did not use the Scheffé test. We have corrected the text to read, "The one-way analysis of variance was performed for continuous variables." on line 225. Also, we mentioned the level of significance used in the study on line 229.

189 performed for categorical variables in the subgroup analysis. 

Line 148 -149 A score of ≤3 was considered “highly health literate” because this represented the number of participants in the two groups. this statement is not clear, what is the connection of the literacy score and the number of participants? 

→ We apologize for the confusion. The sentence has been revised to clarify that "A score of ≤3 was considered 'highly health literate' because it represented participants who had significant knowledge within both groups." on line 185.

Line 179 logistic regression mixed-effect-Is this, ok? I guess you wanted to say mixed-effect logistic model

→ Thank you for pointing this out. We have updated the terminology to "mixed-effect logistic model" to ensure clarity on line 220.

On line 193 the authors state that, “these scores were summed, a lower total score indicated that the respondents were more knowledgeable ,. The researchers may need to reverse the score coding since most readers will associate high score to success. In this case it is in the opposite direction

→ Thank you for this valuable suggestion. We acknowledge that a lower score indicates greater knowledge, which is opposite to typical expectations. We have clarified this in the manuscript to avoid misunderstanding. We have added clarification on line 237: “It is important to note that, unlike typical scoring systems where higher scores reflect better knowledge, in this study, a lower score reflects higher knowledge.”

Line 264- 265, the alternative hypothesis is not clear. Was the alternative hypothesis one sided since the authors state that hat the proportion of participants with high health literacy in Groups 2 and 3 would be higher than that in Group 1 

→ Thank you for pointing this out. The alternative hypothesis was indeed one-sided, as stated, and we have clarified this in the revised text on line 315: “Conversely, the alternative hypothesis posited that the proportion of participants with high health literacy in Groups 2 and 3 would be greater than that in Group 1, which was a one-sided hypothesis.”

line 365…, for the third questionnaire, ….do the authors mean the third survey here or what is the third questionnaire?

→ Thank you for requesting clarification. We used the same questionnaire at three different time points to assess knowledge development. The "third survey is a collection of responses" refers to the third questionnaire. We have revised the text for clarity on line 423.

On line 381 and 383 the authors use the abbreviations MHLW which is not clearly defined.

→ We apologize for this oversight. MHLW stands for “Ministry of Health, Labour and Welfare,” and we have added this definition at the first use on line 439 in the text.

---

## [Decision Letter · Decision Letter 1]

18 Dec 2024

Effects of different educational interventions on cervical cancer knowledge and human papillomavirus vaccination uptake among young women in Japan: preliminary results of a cluster randomized controlled trial

PONE-D-24-35490R1

Dear Dr. Nagasaka,

We’re pleased to inform you that your manuscript has been judged scientifically suitable for publication and will be formally accepted for publication once it meets all outstanding technical requirements.

Kind regards,

Lucy W. Kivuti-Bitok, Ph.D. MHSM,BScN

Academic Editor

PLOS ONE

Additional Editor Comments (optional):

Reviewers' comments:

Reviewer's Responses to Questions

**Comments to the Author**

1. If the authors have adequately addressed your comments raised in a previous round of review and you feel that this manuscript is now acceptable for publication, you may indicate that here to bypass the “Comments to the Author” section, enter your conflict of interest statement in the “Confidential to Editor” section, and submit your "Accept" recommendation.

Reviewer #2: All comments have been addressed

2. Is the manuscript technically sound, and do the data support the conclusions?

Reviewer #2: Yes

3. Has the statistical analysis been performed appropriately and rigorously? 

Reviewer #2: Yes

4. Have the authors made all data underlying the findings in their manuscript fully available?

Reviewer #2: Yes

5. Is the manuscript presented in an intelligible fashion and written in standard English?

Reviewer #2: Yes

6. Review Comments to the Author

Reviewer #2: The authors positively acknowledged the issues raised in the first review and took time to address and gave explanations . They also made clarifications on the issues that were not initially clear . This led to a great improvement on the manuscript that is easier to read by the audience.

7. PLOS authors have the option to publish the peer review history of their article (what does this mean?). If published, this will include your full peer review and any attached files.

Reviewer #2: No

---

## [Editor Report · Acceptance letter]

26 Dec 2024

PONE-D-24-35490R1 

PLOS ONE

Dear Dr. Nagasaka, 

I'm pleased to inform you that your manuscript has been deemed suitable for publication in PLOS ONE. Congratulations! Your manuscript is now being handed over to our production team.

Kind regards, 

on behalf of

Prof Lucy W. Kivuti-Bitok 

Academic Editor

PLOS ONE